# DIFFUSION-BASED IMAGE TRANSLATION USING DISENTANGLED STYLE AND CONTENT REPRESENTATION

**Gihyun Kwon[1], Jong Chul Ye[2,1]**
Department of Bio and Brain Engineering[1], Kim Jaechul Graduate School of AI[2], KAIST
`cyclomon,jong.ye@kaist.ac.kr`

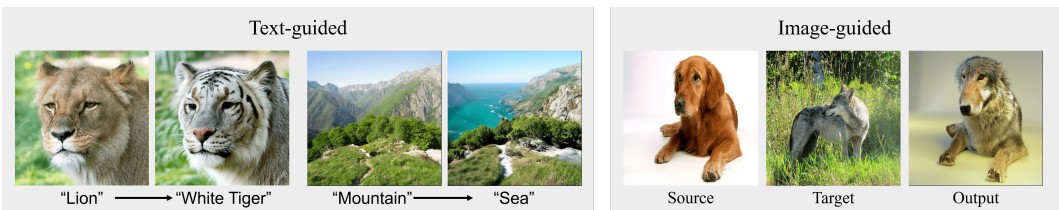

Figure 1: Image translation results by DiffuseIT. Our model can generate high-quality translation outputs using both text and image conditions. More results can be found in the experiment section.

## ABSTRACT

Diffusion-based image translation guided by semantic texts or a single target image has enabled flexible style transfer which is not limited to the specific domains. Unfortunately, due to the stochastic nature of diffusion models, it is often difficult to maintain the original content of the image during the reverse diffusion. To address this, here we present a novel diffusion-based unsupervised image translation method, dubbed as *DiffuseIT*, using disentangled style and content representation. Specifically, inspired by the slicing Vision Transformer (Tumanyan et al., 2022), we extract intermediate keys of multihead self attention layer from ViT model and used them as the content preservation loss. Then, an image guided style transfer is performed by matching the [CLS] classification token from the denoised samples and target image, whereas additional CLIP loss is used for the text-driven style transfer. To further accelerate the semantic change during the reverse diffusion, we also propose a novel semantic divergence loss and resampling strategy. Our experimental results show that the proposed method outperforms state-of-the-art baseline models in both text-guided and image-guided translation tasks.

## 1 INTRODUCTION

Image translation is a task in which the model receives an input image and converts it into a target domain. Early image translation approaches (Zhu et al., 2017; Park et al., 2020; Isola et al., 2017) were mainly designed for single domain translation, but soon extended to multi-domain translation (Choi et al., 2018; Lee et al., 2019). As these methods demand large training set for each domain, image translation approaches using only a single image pairs have been studied, which include the one-to-one image translation using multiscale training (Lin et al., 2020), or patch matching strategy (Granot et al., 2022; Kolkin et al., 2019). Most recently, Splicing ViT (Tumanyan et al., 2022) exploits a pre-trained DINO ViT (Caron et al., 2021) to convert the semantic appearance of a given image into a target domain while maintaining the structure of input image.

On the other hand, by employing the recent text-to-image embedding model such as CLIP (Radford et al., 2021), several approaches have attempted to generate images conditioned on text prompts (Patashnik et al., 2021; Gal et al., 2021; Crowson et al., 2022; Couairon et al., 2022). As these methods rely on Generative Adversarial Networks (GAN) as a backbone generative model, the semantic changes are not often properly controlled when applied to an out-of-data (OOD) image generation.

Recently, score-based generative models (Ho et al., 2020; Song et al., 2020b; Nichol & Dhariwal, 2021) have demonstrated state-of-the-art performance in text-conditioned image generation (Ramesh et al., 2022; Saharia et al., 2022b; Crowson, 2022; Avrahami et al., 2022). However, when it comes to the image translation scenario in which multiple conditions (e.g. input image, text condition) are given to the score based model, disentangling and separately controlling the components still remains as an open problem.

In fact, one of the most important open questions in image translation by diffusion models is to transform only the semantic information (or style) while maintaining the structure information (or content) of the input image. Although this could not be an issue with the conditional diffusion models trained with matched input and target domain images (Saharia et al., 2022a), such training is impractical in many image translation tasks (e.g. summer-to-winter, horse-to-zebra translation). On the other hand, existing methods using unconditional diffusion models often fail to preserve content information due to the entanglement problems in which semantic and content change at the same time (Avrahami et al., 2022; Crowson, 2022). DiffusionCLIP (Kim et al., 2022) tried to address this problem using denoising diffusion implicit models (DDIM) (Song et al., 2020a) and pixel-wise loss, but the score function needs to be fine-tuned for a novel target domain, which is computationally expensive.

In order to control the diffusion process in such a way that it produces the output that simultaneously retain the content of the input image and follow the semantics of the target text or image, here we introduce a loss function using a pre-trained Vision Transformer (ViT) (Dosovitskiy et al., 2020). Specifically, inspired by the recent idea (Tumanyan et al., 2022), we extract intermediate keys of multihead self attention layer and [CLS] classification tokens of the last layer from the DINO ViT model and used them as our content and style regularization, respectively. More specifically, to preserve the structural information, we use the similarity and contrastive loss between intermediate keys of the input and denoised image during the sampling. Then, an image guided style transfer is performed by matching the [CLS] token between the denoised sample and the target domain, whereas additional CLIP loss is used for the text-driven style transfer. To further improve the sampling speed, we propose a novel semantic divergence loss and resampling strategy.

Extensive experimental results including Fig. 1 confirmed that our method provide state-of-the-art performance in both text- and image- guided style transfer tasks quantitatively and qualitatively. To our best knowledge, this is the first unconditional diffusion model-based image translation method that allows both text- and image- guided style transfer without altering input image content.

## 2 RELATED WORK

**Text-guided image synthesis.** Thanks to the outstanding performance of text-to-image alignment in the feature space, CLIP has been widely used in various text-related computer vision tasks including object generation (Liu et al., 2021; Wang et al., 2022a), style transfer (Kwon & Ye, 2021; Fu et al., 2021), object segmentation (Lüddecke & Ecker, 2022; Wang et al., 2022b), etc. Several recent approaches also demonstrated state-of-the-art performance in text-guided image manipulation task by combining the CLIP with image generation models. Previous approaches leverage pre-trained StyleGAN (Karras et al., 2020) for image manipulation with a text condition (Patashnik et al., 2021; Gal et al., 2021; Wei et al., 2022). However, StyleGAN-based methods cannot be used in arbitrary natural images since it is restricted to the pre-trained data domain. Pre-trained VQGAN (Esser et al., 2021) was proposed for better generalization capability in the image manipulation, but it often suffers from poor image quality due to limited power of the backbone model.

With the advance of score-based generative models such as Denoising Diffusion Probabilistic Model (DDPM) (Ho et al., 2020), several methods (Ramesh et al., 2022; Saharia et al., 2022b) tried to generate photo-realistic image samples with given text conditions. However, these approaches are not adequate for image translation framework as the text condition and input image are not usually disentangled. Although DiffusionCLIP (Kim et al., 2022) partially solves the problem using DDIM sampling and pixelwise regularization during the reverse diffusion, it has major disadvantage in that it requires fine-tuning process of score models. As a concurrent work, DDIB(Su et al., 2022) proposed diffusion model based image translation using deterministic probability flow ODE formulation.

**Single-shot Image Translation.** In image translation using single target image, early models mainly focused on image style transfer (Gatys et al., 2016; Huang & Belongie, 2017; Park & Lee, 2019; Yoo et al., 2019). Afterwards, methods using StyleGAN adaptation (Ojha et al., 2021; Zhu et al., 2021; Kwon & Ye, 2022; Chong & Forsyth, 2021) showed great performance, but there are limitations as the models are domain-specific (e.g. human faces). In order to overcome this, methods for converting unseen image into a semantic of target (Lin et al., 2020; Kolkin et al., 2019; Granot et al., 2022) have been proposed, but these methods often suffer from degraded image quality. Recently, Splicing ViT (Tumanyan et al., 2022) successfully exploited pre-trained DINO ViT(Caron et al., 2021) to convert the semantic appearance of given image into target domain while preserving the structure of input.

## 3 PROPOSED METHOD

### 3.1 DDPM SAMPLING WITH MANIFOLD CONSTRAINT

In DDPMs (Ho et al., 2020), starting from a clean image $\boldsymbol{x}_0 \sim q(\boldsymbol{x}_0)$, a forward diffusion process $q(\boldsymbol{x}_t|\boldsymbol{x}_{t-1})$ is described as a Markov chain that gradually adds Gaussian noise at every time steps $t$:

$$q(\boldsymbol{x}_T|\boldsymbol{x}_0) := \prod_{t=1}^{T} q(\boldsymbol{x}_t|\boldsymbol{x}_{t-1}), \quad \text{where} \quad q(\boldsymbol{x}_t|\boldsymbol{x}_{t-1}) := \mathcal{N}(\boldsymbol{x}_t; \sqrt{1-\beta_t}\boldsymbol{x}_{t-1}, \beta_t \boldsymbol{I}), \quad (1)$$

where $\{\beta\}_{t=0}^{T}$ is a variance schedule. By denoting $\alpha_t := 1 - \beta_t$ and $\bar{\alpha}_t := \prod_{s=1}^{t} \alpha_s$, the forward diffused sample at $t$, i.e. $\boldsymbol{x}_t$, can be sampled in one step as:

$$\boldsymbol{x}_t = \sqrt{\bar{\alpha}_t}\boldsymbol{x}_0 + \sqrt{1-\bar{\alpha}_t}\boldsymbol{\epsilon}, \quad \text{where} \quad \boldsymbol{\epsilon} \sim \mathcal{N}(\boldsymbol{0}, \boldsymbol{I}). \quad (2)$$

As the reverse of the forward step $q(\boldsymbol{x}_{t-1}|\boldsymbol{x}_t)$ is intractable, DDPM learns to maximize the variational lowerbound through a parameterized Gaussian transitions $p_\theta(\boldsymbol{x}_{t-1}|\boldsymbol{x}_t)$ with the parameter $\theta$. Accordingly, the reverse process is approximated as Markov chain with learned mean and fixed variance, starting from $p(\boldsymbol{x}_T) = \mathcal{N}(\boldsymbol{x}_T; \boldsymbol{0}, \boldsymbol{I})$:

$$p_\theta(\boldsymbol{x}_{0:T}) := p_\theta(\boldsymbol{x}_T) \prod_{t=1}^{T} p_\theta(\boldsymbol{x}_{t-1}|\boldsymbol{x}_t), \quad \text{where} \quad p_\theta(\boldsymbol{x}_{t-1}|\boldsymbol{x}_t) := \mathcal{N}(\boldsymbol{x}_{t-1}; \boldsymbol{\mu}_\theta(\boldsymbol{x}_t, t), \sigma_t^2 \boldsymbol{I}). \quad (3)$$

where

$$\boldsymbol{\mu}_\theta(\boldsymbol{x}_t, t) := \frac{1}{\sqrt{\alpha_t}}\left(\boldsymbol{x}_t - \frac{1-\alpha_t}{\sqrt{1-\bar{\alpha}_t}}\boldsymbol{\epsilon}_\theta(\boldsymbol{x}_t, t)\right), \quad (4)$$

Here, $\boldsymbol{\epsilon}_\theta(\boldsymbol{x}_t, t)$ is the diffusion model trained by optimizing the objective:

$$\min_\theta L(\theta), \quad \text{where} \quad L(\theta) := \mathbb{E}_{t,\boldsymbol{x}_0,\boldsymbol{\epsilon}}\left[\|\boldsymbol{\epsilon} - \boldsymbol{\epsilon}_\theta(\sqrt{\bar{\alpha}_t}\boldsymbol{x}_0 + \sqrt{1-\bar{\alpha}_t}\boldsymbol{\epsilon}, t)\|^2\right]. \quad (5)$$

After the optimization, by plugging learned score function into the generative (or reverse) diffusion process, one can simply sample from $p_\theta(\boldsymbol{x}_{t-1}|\boldsymbol{x}_t)$ by

$$\boldsymbol{x}_{t-1} = \boldsymbol{\mu}_\theta(\boldsymbol{x}_t, t) + \sigma_t \boldsymbol{\epsilon} = \frac{1}{\sqrt{\alpha_t}}\left(\boldsymbol{x}_t - \frac{1-\alpha_t}{\sqrt{1-\bar{\alpha}_t}}\boldsymbol{\epsilon}_\theta(\boldsymbol{x}_t, t)\right) + \sigma_t \boldsymbol{\epsilon} \quad (6)$$

In image translation using *conditional* diffusion models (Saharia et al., 2022a; Sasaki et al., 2021), the diffusion model $\boldsymbol{\epsilon}_\theta$ in (5) and (6) should be replaced with $\boldsymbol{\epsilon}_\theta(\boldsymbol{y}, \sqrt{\bar{\alpha}_t}\boldsymbol{x}_0 + \sqrt{1-\bar{\alpha}_t}\boldsymbol{\epsilon}, t)$ where $\boldsymbol{y}$ denotes the matched target image. Accordingly, the sample generation is tightly controlled by the matched target in a supervised manner, so that the image content change rarely happen. Unfortunately, the requirement of the *matched* targets for the training makes this approach impractical.

To address this, Dhariwal & Nichol (2021) proposed classifier-guided image translation using the unconditional diffusion model training as in (5) and a pre-trained classifier $p_\phi(\boldsymbol{y}|\boldsymbol{x}_t)$. Specifically, $\boldsymbol{\mu}_\theta(\boldsymbol{x}_t, t)$ in (4) and (6) are supplemented with the gradient of the classifier, i.e. $\hat{\mu}_\theta(\boldsymbol{x}_t, t) := \boldsymbol{\mu}_\theta(\boldsymbol{x}_t, t) + \sigma_t \nabla_{\boldsymbol{x}_t} \log p_\phi(\boldsymbol{y}|\boldsymbol{x}_t)$. However, most of the classifiers, which should be separately trained, are not usually sufficient to control the content of the samples from the reverse diffusion process.

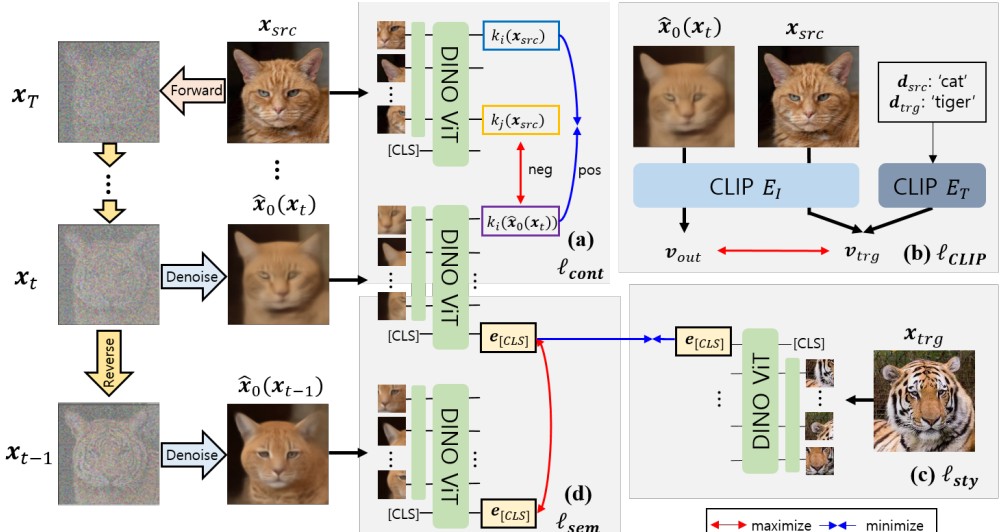

Figure 2: Given the input image $\boldsymbol{x}_{src}$, we guide the reverse diffusion process $\{\boldsymbol{x}_t\}_{t=T}^0$ using various losses. (a) $\ell_{cont}$: the structural similarity loss between input and outputs in terms of contrastive loss between extracted keys from ViT. (b) $\ell_{CLIP}$: relative distance to the target text $\boldsymbol{d}_{trg}$ in CLIP space in terms of $\boldsymbol{x}_{src}$ and $\boldsymbol{d}_{src}$. (c) $\ell_{sty}$: the [CLS] token distances between the outputs and target $\boldsymbol{x}_{trg}$. (d) $\ell_{sem}$: dissimilarity between the [CLS] token from the present and past denoised samples.

Inspired by the recent manifold constrained gradient (MCG) for inverse problems (Chung et al., 2022a), here we formulate our content and style guidance problem as an inverse problem, which can be solved by minimizing the following total cost function with respect to the sample $\boldsymbol{x}$:

$$\ell_{total}(\boldsymbol{x}; \boldsymbol{x}_{trg}, \boldsymbol{x}_{src}), \quad \text{or} \quad \ell_{total}(\boldsymbol{x}; \boldsymbol{d}_{trg}, \boldsymbol{x}_{src}, \boldsymbol{d}_{src}) \tag{7}$$

where $\boldsymbol{x}_{src}$ and $\boldsymbol{x}_{trg}$ refer to the source and target images, respectively; and $\boldsymbol{d}_{src}$ and $\boldsymbol{d}_{trg}$ refer to the source and target text, respectively. In our paper, the first form of the total loss in (7) is used for image-guided translation, where the second form is for the text-guided translation. Then, the sampling from the reverse diffusion with MCG is given by

$$\boldsymbol{x}'_{t-1} = \frac{1}{\sqrt{\alpha_t}}\left(\boldsymbol{x}_t - \frac{1-\alpha_t}{\sqrt{1-\bar{\alpha}_t}}\boldsymbol{\epsilon}_\theta(\boldsymbol{x}_t, t)\right) + \sigma_t\boldsymbol{\epsilon} \tag{8}$$

$$\boldsymbol{x}_{t-1} = \boldsymbol{x}'_{t-1} - \nabla_{\boldsymbol{x}_t}\ell_{total}(\hat{\boldsymbol{x}}_0(\boldsymbol{x}_t)) \tag{9}$$

where $\hat{\boldsymbol{x}}_0(\boldsymbol{x}_t)$ refers to the estimated clean image from the sample $\boldsymbol{x}_t$ using the Tweedie's formula (Kim & Ye, 2021):

$$\hat{\boldsymbol{x}}_0(\boldsymbol{x}_t) := \frac{\boldsymbol{x}_t}{\sqrt{\bar{\alpha}_t}} - \frac{\sqrt{1-\bar{\alpha}_t}}{\sqrt{\bar{\alpha}_t}}\boldsymbol{\epsilon}_\theta(\boldsymbol{x}_t, t). \tag{10}$$

In the following, we describe how the total loss $\ell_{total}$ is defined. For brevity, we notate $\hat{\boldsymbol{x}}_0(\boldsymbol{x}_t)$ as $\boldsymbol{x}$ in the following sections.

### 3.2 STRUCTURE LOSS

As previously mentioned, the main objective of image translation is maintaining the content structure between output and the input image, while guiding the output to follow semantic of target condition. Existing methods (Couairon et al., 2022; Kim et al., 2022) use pixel-wise loss or the perceptual loss for the content preservation. However, the pixel space does not explicitly discriminate content and semantic components: too strong pixel loss hinders the semantic change of output, whereas weak pixel loss alters the structural component along with semantic changes. To address the problem, we need to separately process the semantic and structure information of the image.

Recently, (Tumanyan et al., 2022) demonstrated successful disentanglement of both components using a pre-trained DINO ViT (Caron et al., 2021). They showed that in ViT, the keys $k^l$ of multi-head

self attention (MSA) layer contain structure information, and [CLS] token of last layer contains the semantic information. With above features, they proposed a loss for maintaining structure between input and network output with matching the self similarity matrix $S^l$ of the keys, which can be represented in the following form for our problem:

$$\ell_{ssim}(\boldsymbol{x}_{src}, \boldsymbol{x}) = \|S^l(\boldsymbol{x}_{src}) - S^l(\boldsymbol{x})\|_F, \quad \text{where} \quad \left[S^l(\boldsymbol{x})\right]_{i,j} = \cos(k_i^l(\boldsymbol{x}), k_j^l(\boldsymbol{x})), \quad (11)$$

where $k_i^l(\boldsymbol{x})$ and $k_j^l(\boldsymbol{x})$ indicate $i, j$th key in the $l$-th MSA layer extracted from ViT with image $\boldsymbol{x}$. The self-similarity loss can maintain the content information between input and output, but we found that only using this loss results in a weak regularization in our DDPM framework. Since the key $k_i$ contains the spatial information corresponding the $i$-th patch location, we use additional regularization with contrastive learning as shown in Fig. 2(a), inspired by the idea of using both of relation consistency and contrastive learning(Jung et al., 2022). Specifically, leveraging the idea of patch contrastive loss (Park et al., 2020), we define the infoNCE loss using the DINO ViT keys:

$$\ell_{cont}(\boldsymbol{x}_{src}, \boldsymbol{x}) = -\sum_i \log \left( \frac{\exp(\text{sim}(k_i^l(\boldsymbol{x}), k_i^l(\boldsymbol{x}_{src}))/\tau)}{\exp(\text{sim}(k_i^l(\boldsymbol{x}), k_i^l(\boldsymbol{x}_{src}))/\tau + \sum_{j \neq i} \exp(\text{sim}(k_i^l(\boldsymbol{x}), k_j^l(\boldsymbol{x}_{src}))/\tau)} \right),$$
$$(12)$$

where $\tau$ is temperature, and $\text{sim}(\cdot, \cdot)$ represents the normalized cosine similarity. With this loss, we regularize the key of same positions to have closer distance, while maximizing the distances between the keys at different positions.

### 3.3 STYLE LOSS

**CLIP Loss for Text-guided Image Translation**   Based on the previous work of (Dhariwal & Nichol, 2021), CLIP-guided diffusion (Crowson, 2022) proposed to guide the reverse diffusion using pre-trained CLIP model using the following loss function:

$$\ell_{CLIP}(\boldsymbol{d}_{trg}, \boldsymbol{x}) := -\text{sim}\left(E_T(\boldsymbol{d}_{trg}), E_I(\boldsymbol{x})\right), \quad (13)$$

where $\boldsymbol{d}_{trg}$ is the target text prompt, and $E_I, E_T$ refer to the image and text encoder of CLIP, respectively. Although this loss can give text-guidance to diffusion model, the results often suffer from poor image quality.

Instead, we propose to use input-aware directional CLIP loss (Gal et al. (2021)) which matches the CLIP embedding of the output image to the target vector in terms of $\boldsymbol{d}_{trg}$, $\boldsymbol{d}_{src}$, and $\boldsymbol{x}_{src}$. More specifically, our CLIP-based semantic loss is described as (see also Fig. 2(b)):

$$\ell_{CLIP}(\boldsymbol{x}; \boldsymbol{d}_{trg}, \boldsymbol{x}_{src}, \boldsymbol{d}_{src}) := -\text{sim}(\boldsymbol{v}_{trg}, \boldsymbol{v}_{src}) \quad (14)$$

where

$$\boldsymbol{v}_{trg} := E_T(\boldsymbol{d}_{trg}) + \lambda_i E_I(\boldsymbol{x}_{src}) - \lambda_s E_T(\boldsymbol{d}_{src}), \quad \boldsymbol{v}_{src} := E_I(\text{aug}(\boldsymbol{x})) \quad (15)$$

where $\text{aug}(\cdot)$ denotes the augmentation for preventing adversarial artifacts from CLIP. Here, we simultaneously remove the source domain information $-\lambda_s E_T(\boldsymbol{d}_{src})$ and reflect the source image information to output $+\lambda_i E_I(\boldsymbol{x}_{src})$ according to the values of $\lambda_s$ and $\lambda_i$. Therefore it is possible to obtain stable outputs compared to using the conventional loss.

Furthermore, in contrast to the existing methods using only single pre-trained CLIP model (e.g. ViT/B-32), we improve the text-image embedding performance by using the recently proposed CLIP model ensemble method (Couairon et al. (2022)). Specifically, instead of using a single embedding, we concatenate the multiple embedding vectors from multiple pre-trained CLIP models and used the it as our final embedding.

**Semantic Style Loss for Image-guided Image Translation**   In the case of image-guide translation, we propose to use [CLS] token of ViT as our style guidance. As explained in the previous part 3.2, the [CLS] token contains the semantic style information of the image. Therefore, we can guide the diffusion process to match the semantic of the samples to that of target image by minimizing the [CLS] token distances as shown in Fig. 2(c). Also, we found that using only [CLS] tokens often results in misaligned color values. To prevent this, we guide the output to follow the overall color statistic of target image with weak MSE loss between the images. Therefore, our loss function is described as follows:

$$\ell_{sty}(\boldsymbol{x}_{trg}, \boldsymbol{x}) = ||\boldsymbol{e}_{[CLS]}^L(\boldsymbol{x}_{trg}) - \boldsymbol{e}_{[CLS]}^L(\boldsymbol{x})||_2 + \lambda_{mse}||\boldsymbol{x}_{trg} - \boldsymbol{x}||_2. \quad (16)$$

where $\boldsymbol{e}_{[CLS]}^L$ denotes the last layer [CLS] token.

### 3.4 Acceleration Strategy

**Semantic Divergence Loss**   With the proposed loss functions, we can achieve text- or image-guided image translation outputs. However, we empirically observed that the generation process requires large steps to reach the the desired output. To solve the problem, we propose a simple approach to accelerate the diffusion process. As explained before, the [CLS] token of ViT contains the overall semantic information of the image. Since our purpose is to make the semantic information as different from the original as possible while maintaining the structure, we conjecture that we can achieve our desired purpose by maximizing the distance between the [CLS] tokens of the previous step and the current output during the generation process as described in Fig. 2(d). Therefore, our loss function at time $t$ is given by

$$\ell_{sem}(\boldsymbol{x}_t; \boldsymbol{x}_{t+1}) = -||\boldsymbol{e}^L_{[CLS]}(\hat{\boldsymbol{x}}_0(\boldsymbol{x}_t)) - \boldsymbol{e}^L_{[CLS]}(\hat{\boldsymbol{x}}_0(\boldsymbol{x}_{t+1}))||_2, \tag{17}$$

Specifically, we maximize the distance between the denoised output of the present time and the previous time, so that next step sample has different semantic from the previous step. One could think of alternatives to maximize pixel-wise or perceptual distance, but we have experimentally found that in these cases, the content structure is greatly harmed. In contrast, our proposed loss has advantages in terms of image quality because it can control only the semantic appearance.

**Resampling Strategy**   As shown in CCDF acceleration strategy (Chung et al., 2022b), a better initialization leads to an accelerated reverse diffusion for inverse problem. Empirically, in our image translation problem we also find that finding the good starting point at time step $T$ for the reverse diffusion affects the overall image quality. Specifically, in order to guide the initial estimate $\boldsymbol{x}_T$ to be sufficiently good, we perform $N$ repetition of one reverse sampling $\boldsymbol{x}_{T-1}$ followed by one forward step $\boldsymbol{x}_T = \sqrt{1 - \beta_{T-1}}\boldsymbol{x}_{T-1} + \beta_{T-1}\boldsymbol{\epsilon}$ to find the $\boldsymbol{x}_T$ whose gradient for the next step is easily affected by the loss. With this initial resampling strategy, we can empirically found the initial $\boldsymbol{x}_T$ that can reduce the number of reverse steps. The overall process is in our algorithm in Appendix.

### 3.5 Total Loss

Putting all together, the final loss in (7) for the text-guided reverse diffusion is given by

$$\ell_{total} = \lambda_1 \ell_{cont} + \lambda_2 \ell_{ssim} + \lambda_3 \ell_{CLIP} + \lambda_4 \ell_{sem} + \lambda_5 \ell_{rng}, \tag{18}$$

where $\ell_{rng}$ is a regularization loss to prevent the irregular step of reverse diffusion process suggested in (Crowson (2022)). If the target style image $\boldsymbol{x}_{trg}$ is given instead of text conditions $\boldsymbol{d}_{src}$ and $\boldsymbol{d}_{trg}$, then $\ell_{CLIP}$ is simply substituted for $\ell_{sty}$.

## 4 Experiment

### 4.1 Experimental Details

For implementation, we refer to the official source code of blended diffusion (Avrahami et al. (2022)). All experiments were performed using unconditional score model pre-trained with Imagenet $256\times256$ resolution datasets (Dhariwal & Nichol (2021)). In all the experiments, we used diffusion step of $T = 60$ and the resampling repetition of $N = 10$; therefore, the total of 70 diffusion reverse steps are used. The generation process takes 40 seconds per image in single RTX 3090 unit. In $\ell_{CLIP}$, we used the ensemble of 5 pre-trained CLIP models (RN50, RN50x4, ViT-B/32, RN50x16, ViT-B/16) for the text-guidance, following the setup of Couairon et al. (2022). Our detailed experimental settings are elaborated in Appendix.

### 4.2 Text-guided Semantic Image Translation

To evaluate the performance of our text-guided image translation, we conducted comparisons with state-of-the-art baseline models. For baseline methods, we selected the recently proposed models which use pre-trained CLIP for text-guided image manipulation: VQGAN-CLIP (Crowson et al. (2022)), CLIP-guided diffusion (CGD) (Crowson (2022)), DiffusionCLIP (Kim et al. (2022)), and FlexIT (Couairon et al. (2022)). For all baseline methods, we referenced the official source codes.

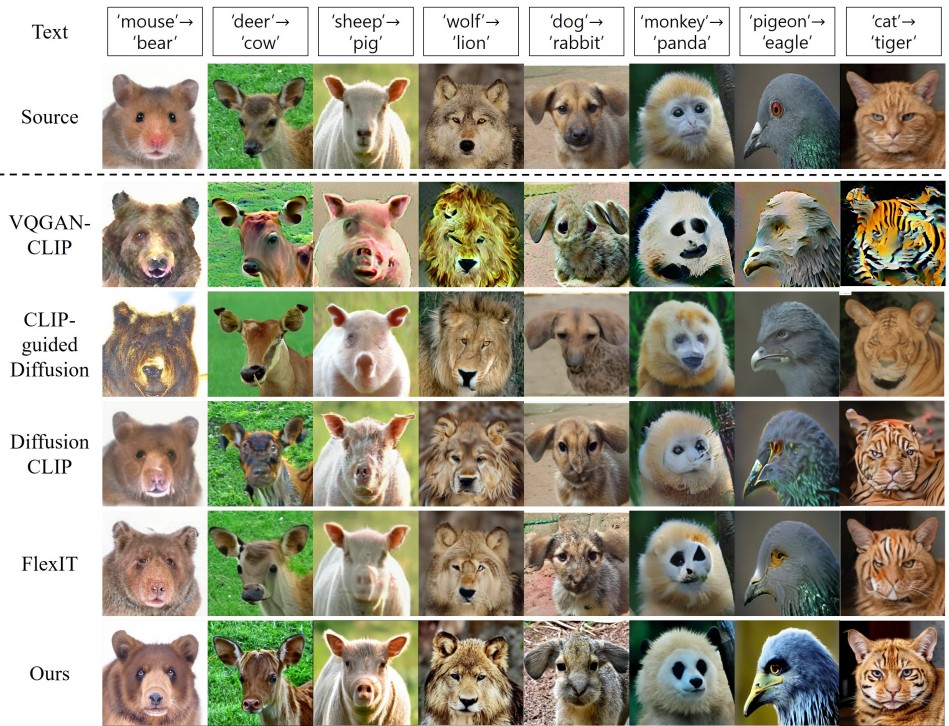

Figure 3: Qualitative comparison of text-guided translation on *Animals* dataset. Our model generates realistic samples that reflects the text condition, with better perceptual quality than the baselines.

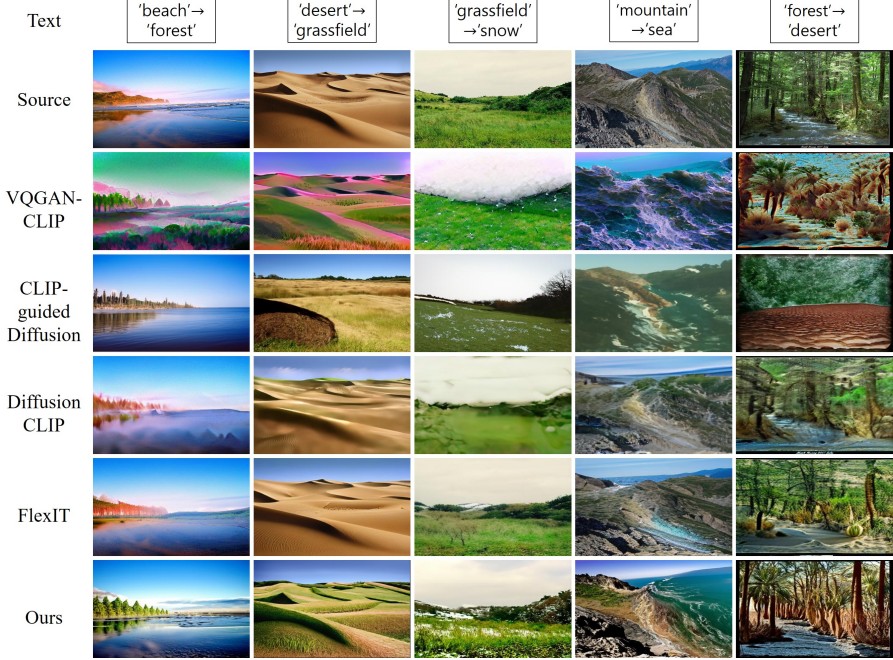

Figure 4: Qualitative comparison of text-guided image translation on *Landscape* dataset. Our model generates outputs with better perceptual quality than the baselines.

Since our framework can be applied to arbitrary text semantics, we tried quantitative and qualitative evaluation on various kinds of natural image datasets. We tested our translation performance using two different datasets: animal faces (Si & Zhu (2012)) and landscapes (Chen et al. (2018)). The animal face dataset contains 14 classes of animal face images, and the landscapes dataset consists of 7 classes of various natural landscape images.

| Method | Animals | | | Landscapes | | | User Study | | |
|---|---|---|---|---|---|---|---|---|---|
| | SFID↓ | CSFID↓ | LPIPS↓ | SFID↓ | CSFID↓ | LPIPS↓ | Text↑ | Realism↑ | Content↑ |
| VQGAN-CLIP | 30.01 | 65.51 | 0.462 | 33.31 | 82.92 | 0.571 | 2.78 | 2.05 | 2.16 |
| CLIP-GD | 12.50 | 53.05 | 0.468 | 18.13 | 62.19 | 0.458 | 2.61 | 2.24 | 2.28 |
| DiffusionCLIP | 25.09 | 66.50 | 0.379 | 29.85 | 76.29 | 0.568 | 2.50 | 2.54 | 3.06 |
| FlexIT | 32.71 | 57.87 | 0.215 | 18.04 | 60.04 | 0.243 | 2.22 | 3.15 | 3.89 |
| Ours | 9.98 | 41.07 | 0.372 | 16.86 | 54.48 | 0.417 | 3.68 | 4.28 | 4.11 |

Table 1: Quantitative comparison in the text-guided image translation. Our model outperforms baselines in overall scores for both of *Animals* and *Landscapes* datasets as well as user study.

To measure the performance of the generated images, we measured the FID score (Heusel et al. (2017)). However, when using the basic FID score measurement, the output value is not stable because our number of generated images is not large. To compensate for this, we measure the performance using a simplified FID (Kim et al. (2020)) that does not consider the diagonal term of the feature distributions. Also, we additionally showed a class-wise SFID score that measures the SFID for each class of the converted output because it is necessary to measure whether the converted output accurately reflects the semantic information of the target class. Finally, we used the averaged LPIPS score between input and output to verify the content preservation performance of our method. Further experimental settings can be found in our Appendix.

In Table 1, we show the quantitative comparison results. In image quality measurement using SFID and CSFID, our model showed the best performance among all baseline methods. Especially for Animals dataset, our SFID value outperformed others in large gain. In the content preservation by LPIPS score, our method scored the second best. In case of FlexIT, it showed the best score in LPIPS since the model is directly trained with LPIPS loss. However, too low value of LPIPS is undesired as it means that the model failed in proper semantic change. This can be also seen in qualitative result of Figs. 3 and 4, where our results have proper semantic features of target texts with content preservation, whereas the results from FlexIT failed in semantic change as it is too strongly confined to the source images. In other baseline methods, most of the methods failed in proper content preservation. Since our method is based on DDPM, our model can generate diverse images as shown in the additional outputs in our Appendix.

To further evaluate the perceptual quality of generated samples, we conducted user study. In order to measure the detailed opinions, we used custom-made opinion scoring system. We asked the users in three different parts: 1) Are the output have correct semantic of target text? (Text-match), 2) are the generated images realistic? (Realism), 3) do the outputs contains the content information of source images? (Content). Detailed user-study settings are in our Appendix. In Table 1, our model showed the best performance, which further shows the superiority of our method.

### 4.3 IMAGE-GUIDED SEMANTIC IMAGE TRANSLATION

Since our method can be easily adapted to the image translation guided by target images, we evaluate the performance of our model with comparison experiments. We compare our model with appearance transfer models of Splicing ViT (Tumanyan et al. (2022)), STROTSS (Kolkin et al. (2019)), and style transfer methods WCT2 (Yoo et al. (2019)) and SANet (Park & Lee (2019)).

Fig. 5 is a qualitative comparison result of image guided translation task. Our model successfully generated outputs that follow the semantic styles of the target images while maintaining the content of the source images. In

| Method | Style↑ | Realism↑ | Content↑ |
|---|---|---|---|
| SANet | 2.75 | 4.08 | 4.37 |
| WCT2 | 2.59 | 4.64 | 4.90 |
| STROTSS | 3.92 | 2.91 | 3.17 |
| SplicingViT | 3.50 | 2.08 | 2.15 |
| Ours | 4.23 | 4.25 | 4.51 |

Table 2: User study comparison of image-guided translation tasks. Our model outperforms baseline methods in overall perceptual quality.

the case of other models, we can see that the content was severely deformed or the semantic style was not properly reflected. We also measured the overall perceptual quality through a user study. As with text-guided translation, we investigated user opinion through three different questions. In Table 2, our model obtained the best score in style matching score and the second best in realism and content preservation scores. Baseline WCT2 showed the best in realism and content scores, but it shows the worst score in style matching because the outputs are hardly changed from the inputs except for overall colors. The opinions scores confirm that our model outperforms the baselines. More details are in our Appendix.

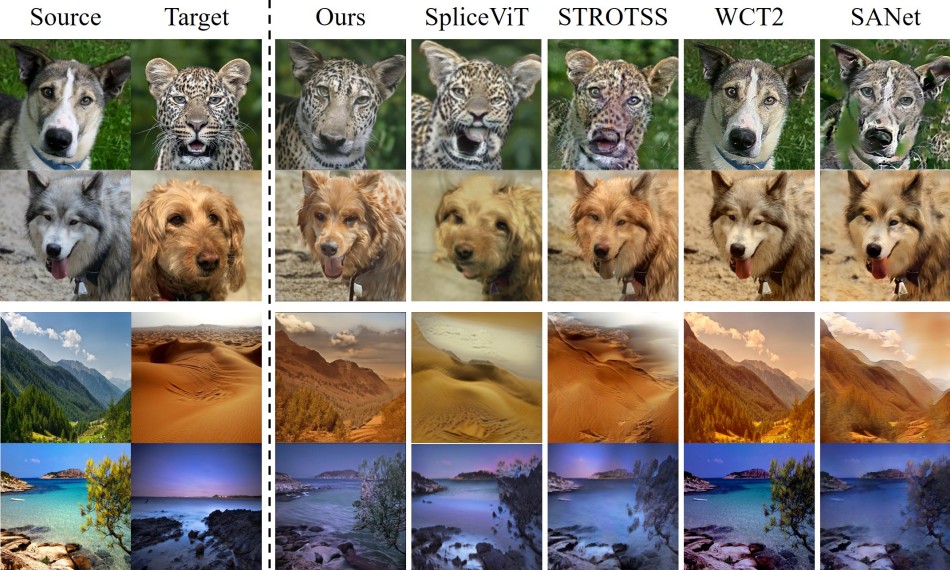

Figure 5: Qualitative comparison of image-guided image translation. Our results have better perceptual quality than the baseline outputs.

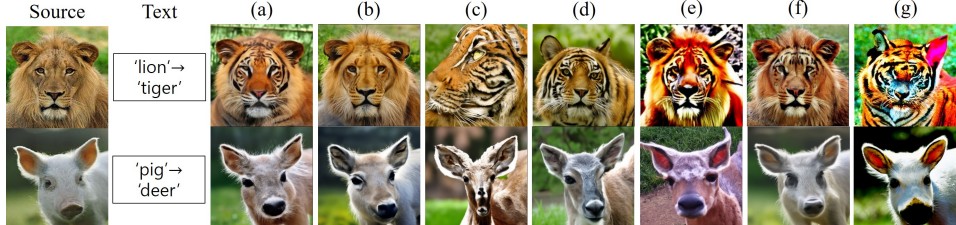

Figure 6: Qualitative comparison on ablation study. Our full setting shows the best results.

## 4.4 ABLATION STUDY

To verify the proposed components in our framework, we compare the generation performance with different settings. In Fig. 6, we show that (a) the outputs from our best setting have the correct semantic of target text, with preserving the content of the source; (b) by removing $\ell_{sem}$, the results still have the appearance of source images, suggesting that images are not fully converted to the target domain; (c) without $\ell_{cont}$, the output images totally failed to capture the content of source images; (d) by using LPIPS perceptual loss instead of proposed $\ell_{cont}$, the results can only capture the approximate content of source images; (e) using pixel-wise $l_2$ maximization loss instead of proposed $\ell_{sem}$, the outputs suffer from irregular artifacts; (f) without using our proposed resampling trick, the results cannot fully reflect the semantic information of target texts. (g) With using VGG16 network instead of DINO ViT, the output structure is severely degraded with artifacts. Overall, we can obtain the best generation outputs by using all of our proposed components. For further evaluation, we will show the quantitative results of ablation study in our Appendix.

## 5 CONCLUSION

In conclusion, we proposed a novel loss function which utilizes a pre-trained ViT model to guide the generation process of DDPM models in terms of content preservation and semantic changes. We further propose a novel strategy of resampling technique for better initialization of diffusion process. For evaluation, our extensive experimental results show that our proposed framework has superior performance compared to baselines in both of text- and image-guided semantic image translation tasks. Despite the successful results, our method often fails to translate the image styles when there is large domain gap between source and target. With respect to this, we show the failure cases and discussions on limitations in Appendix.

## Acknowledgement

This research was supported by Field-oriented Technology Development Project for Customs Administration through National Research Foundation of Korea(NRF) funded by the Ministry of Science & ICT and Korea Customs Service(NRF-2021M3I1A1097938), the KAIST Key Research Institute (Interdisciplinary Research Group) Project, and the National Research Foundation of Korea under Grant (NRF-2020R1A2B5B03001980).

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

## A    EXPERIMENTAL DETAILS

### A.1    IMPLEMENTATION DETAILS

For implementation, in case of using text-guided image manipulation, our initial sampling numbers are set as $T = 100$, but we skipped the initial 40 steps to maintain the abstract content of input image. Therefore, the total number of sampling steps is $T = 60$. With resampling step of $N = 10$, we use total of 70 iterations for single image output. We found that using more resampling steps does not show meaningful performance improvement. In image-guided manipulation, we set initial sampling number $T = 200$, and skipped the initial 80 steps. We used resampling step $N = 10$. Therefore, we use total of 130 iterations. Although we used more iterations than text-guided translation, it takes about 40 seconds.

For hyperparameters, we use $\lambda_1 = 200$, $\lambda_2 = 100$, $\lambda_3 = 2000$, $\lambda_4 = 1000$, $\lambda_5 = 200$. For image-guided translation, we set $\lambda_{mse} = 1.5$. For our CLIP loss, we set $\lambda_s = 0.4$ and $\lambda_i = 0.2$. For our ViT backbone model, we used pre-trained DINO ViT that follows the baseline of Splicing ViT (Tumanyan et al., 2022). For extracting keys of intermediate layer, we use layer of $l = 11$, and for [CLS] token, we used last layer output. Since ViT and CLIP model only take 224×224 resolution images, we resized all images before calculating the losses with ViT and CLIP.

To further improve the sample quality of our qualitative results, we used restarting trick in which we check the $\ell_{reg}$ loss calculated at initial time step $T$, and restart the whole process if the loss value is too high. If the initial loss $\ell_{reg} > 0.01$, we restarted the process. For quantitative result, we did not use the restart trick for fair comparison.

For augmentation, we use the same geometrical augmentations proposed in FlexIT(Couairon et al., 2022). Also, following the setting from CLIP-guided diffusion(Crowson, 2022), we included noise augmentation in which we mix the noisy image to $\hat{x}_0(x_t)$ as it further removes the artifacts.

In our image-guided image translation on natural landscape images, we matched the color distribution of output image to that of target image with (Hahne & Aggoun, 2021), as it showed better perceptual quality. Our detailed implementation can be found in our official GitHub repository.[1]

For baseline experiments, we followed the official source codes in all of the models[2345]. For diffusion-based models (DiffusionCLIP, CLIP-guided diffusion), we used unconditional score model pre-trained on 256×256 resolutions. In DiffusionCLIP, we fine-tuned the score model longer than suggested training iteration, as it showed better quality. In CLIP-guided diffusion, we set the CLIP-guided loss as 2000, and also set initial sampling number as $T = 100$ with skipping initial 40 steps. For VQGAN-based models (FlexIT, VQGAN-CLIP), we used VQGAN trained on imagenet 256×256 resolutions datasets. In VQGAN-CLIP, as using longer iteration results in extremely degraded images, therefore we optimized only 30 iterations, which is smaller than suggested iterations ( ≥80). In the experiments of FlexIT, we followed the exactly same settings suggested in the original paper.

For baselines of image-guided style transfer tasks, we also referenced the original source codes[6789]. In all of the experiments, we followed the suggested settings from the original papers.

### A.2    DATASET DETAILS

For our quantitative results using text-guided image translation, we used two different datasets *Animals* and *Landscapes*. In Animals dataset, the original dataset contains 21 different classes, but we filtered out the images from 14 classes (bear, cat, cow, deer, dog, lion, monkey, mouse, panda, pig,

---

[1]`https://github.com/anon294384/DiffuseIT`
[2]`https://github.com/afiaka87/clip-guided-diffusion`
[3]`https://github.com/nerdyrodent/VQGAN-CLIP`
[4]`https://github.com/gwang-kim/DiffusionCLIP`
[5]`https://github.com/facebookresearch/SemanticImageTranslation`
[6]`https://github.com/omerbt/Splice`
[7]`https://github.com/nkolkin13/STROTSS`
[8]`https://github.com/clovaai/WCT2`
[9]`https://github.com/GlebSBrykin/SANET`

rabbit, sheep, tiger, wolf) which can be classified as mammals. Remaining classes (e.g. human, chicken, etc.) are removed since they have far different semantics from the mammal faces.Therefore we reported quantitative scores only with filtered datasets for fair comparison. The dataset contains 100-300 images per each class, and we selected 4 testing images from each class in order to use them as our content source images. With selected samples, we calculated the metrics using the outputs of translating the 4 images from a source class into all the remaining classes. Therefore, in our animal face dataset, total of 676 generated images are used for evaluation.

In *Landscapes* dataset, we manually classified the images into 7 different classes (beach, desert, forest, grass field, mountain, sea, snow). Each class has 300 different images except for desert class which have 100 different images. Since some classes have not enough number of images, we borrowed images from *seasons* (Anoosheh et al., 2018) dataset. For metric calculation, we selected 8 testing images from each class, and used them as our content source images. Again, we translated the 8 images from source class into all the remaining classes. Therefore, a total of 336 generated images are used for our quantitative evaluation.

For single image guided translation, we selected random images from AFHQ dataset for animal face translation; and for natural image generation, we selected random images from our *Landscapes* datasets.

### A.3  USER STUDY DETAILS

For our user study in text-guided image translation task, we generated 130 different images using 13 different text conditions with our proposed and baseline models. Then we randomly selected 65 images and made 6 different questions. More specifically, we asked the participants question about three different parts: 1) Are the outputs have correct semantic of target text? (Text-match), 2) Are the generated images realistic? (Realism), 3) Do the outputs contain the content information of source images (Content). We randomly recruited a total of 30 users, and provided them the questions using Google Form. The 30 different users come from age group 20s and 50s. We set the minimum score as 1, and the maximum score is 5. The users can score among 5 different options : 1-Very Unlikely, 2-Unlikely, 3-Normal, 4-Likely, 5-Very Likely.

For the user study on image-guided translation task, we generated 40 different images using 8 different images conditions. Then we followed the same protocol to user study on text-guided image translation tasks, except for the content of questions. We asked the users in three different parts: 1) Are the outputs have correct semantic of target style image? (Style-match), 2) Are the generated images realistic? (Realism), 3) Do the outputs contain the content information of source images (Content).

### A.4  ALGORITHM

For detailed explanation, we include Algorithm of our proposed image translation mathods in Algorithm 1.

## B  QUANTITATIVE ABLATION STUDY

For more thorough evaluation of our proposed components, we report ablation study on quantitative metrics. In this experiment, we only used *Animals* dataset due to the time limit. In Table 3, we show the quantitative results on various settings. When we remove one of our acceleration strategies, in setting (b) and (f), we can see that the fid score is degraded as the outputs are not properly changed from the original source images. (e) When we use L2 maximization instead of our proposed $\ell_{sem}$, FID scores are improved from setting (b), but still the performance is not on par with our best settings. (d) When we use weak content regularization using LPIPS, we can see that the overall scores are degraded. When we remove our proposed $\ell_{cont}$, we can observe that SFID and CSFID scores are lower than other settings. However, we can see that LPIPS score is severely high as the model hardly reflect the content information of original source images. (g) we use pre-trained VGG instead of using ViT for ablation study. Instead of ViT keys for structure loss, we substitute it with features extracted from VGG16 relu3_1 activation layer. Also, we substitute ViT [CLS] token with VGG16 relu5_1 feature as it contains high-level semantic features. We can see that the model

---

**Algorithm 1** Semantic image translation: given a diffusion score model $\epsilon_\theta(\boldsymbol{x}_t, t)$, CLIP model, and VIT model

---

**Input:** source image $\boldsymbol{x}_{src}$, diffusion steps $T$, resampling steps $N$, target text $\boldsymbol{d}_{trg}$, source text $\boldsymbol{d}_{src}$ or target image $\boldsymbol{x}_{trg}$
**Output:** translated image $\hat{\boldsymbol{x}}$ which has semantic of $\boldsymbol{d}_{trg}$ (or $\boldsymbol{x}_{trg}$) and content of $\boldsymbol{x}_{src}$
$\boldsymbol{x}_T \sim \mathcal{N}(\sqrt{\bar{\alpha}_t}\boldsymbol{x}_{src}, (1 - \bar{\alpha}_t)\mathbf{I})$, index for resampling $n = 0$

 1: **for all** $t$ from $T$ to $0$ **do**
 2:     $\epsilon \leftarrow \boldsymbol{\epsilon}_\theta(\boldsymbol{x}_t, t)$
 3:     $\hat{\boldsymbol{x}}_0(\boldsymbol{x}_t) \leftarrow \frac{\boldsymbol{x}_t}{\sqrt{\bar{\alpha}_t}} - \frac{\sqrt{1 - \bar{\alpha}_t}}{\sqrt{\bar{\alpha}_t}}\epsilon$
 4:     **if** text-guided **then**
 5:         $\nabla_{total} \leftarrow \nabla_{\boldsymbol{x}_t}\ell_{total}(\hat{\boldsymbol{x}}_0(\boldsymbol{x}_t); \boldsymbol{d}_{trg}, \boldsymbol{x}_{src}, \boldsymbol{d}_{src})$
 6:     **else if** image-guided **then**
 7:         $\nabla_{total} \leftarrow \nabla_{\boldsymbol{x}_t}\ell_{total}(\hat{\boldsymbol{x}}_0(\boldsymbol{x}_t); \boldsymbol{x}_{trg}, \boldsymbol{x}_{src})$
 8:     **end if**
 9:     $\boldsymbol{z} \sim \mathcal{N}(0, \mathbf{I})$
10:     $\boldsymbol{x}'_{t-1} = \frac{1}{\sqrt{\alpha_t}}\left(\boldsymbol{x}_t - \frac{1 - \alpha_t}{\sqrt{1 - \bar{\alpha}_t}}\boldsymbol{\epsilon}\right) + \sigma_t \boldsymbol{z}$
11:     $\boldsymbol{x}_{t-1} = \boldsymbol{x}'_{t-1} - \nabla_{total}$
12:     **if** $t = T$ and $n < N$ **then**
13:         $\boldsymbol{x}_t \leftarrow \mathcal{N}(\sqrt{1 - \beta_{t-1}}\boldsymbol{x}_{t-1}, \beta_{t-1}\mathbf{I})$
14:         $n \leftarrow n + 1$
15:         **go to** 2
16:     **end if**
17: **end for**
18: **return** $\boldsymbol{x}_{-1}$

---

| Settings | Animals | | |
| --- | --- | --- | --- |
| | SFID↓ | CSFID↓ | LPIPS↓ |
| VGG instead of ViT (g) | 9.72 | 43.08 | 0.518 |
| No resampling (f) | 11.88 | 59.09 | 0.316 |
| L2 Max instead of $\ell_{sem}$ (e) | 13.18 | 49.47 | 0.324 |
| LPIPs instead of $\ell_{cont}$ (d) | 11.15 | 58.67 | 0.400 |
| No $\ell_{cont}$ (c) | 9.90 | 33.07 | 0.477 |
| No $\ell_{sem}$ (b) | 15.00 | 53.43 | 0.347 |
| Ours (a) | 9.98 | 41.07 | 0.372 |

Table 3: Quantitative comparison of ablation studies.

shows decent SFID and CSFID scores, but the LPIPS score is very high. The result show that using VGG does not properly operate as regularization tool, rather it degrades the generation process with damaging the structural consistency. Overall, when using our best setting, we can obtain the best output considering all of the scores.

## C  ARTISTIC STYLE TRANSFER

With our framework, we can easily adapt our method to artistic style transfer. With simply changing the text conditions, or using artistic paintings as our image conditions, we can obtain the artistic style transfer results as shown in Fig. 7.

## D  FACE IMAGE TRANSLATION

Instead of using score mode pre-trained on Imagenet dataset, we can use pre-trained score model on FFHQ human face dataset. In order to keep the face identity between source and output images, we include $\lambda_{id}\ell_{id}$ which leverage pre-trained face identification model ArcFace(Deng et al., 2019). We calculate identity loss between $\boldsymbol{x}_{src}$ and denoised image $\hat{\boldsymbol{x}}_0(\boldsymbol{x}_t)$. We use $\lambda_{id} = 100$.

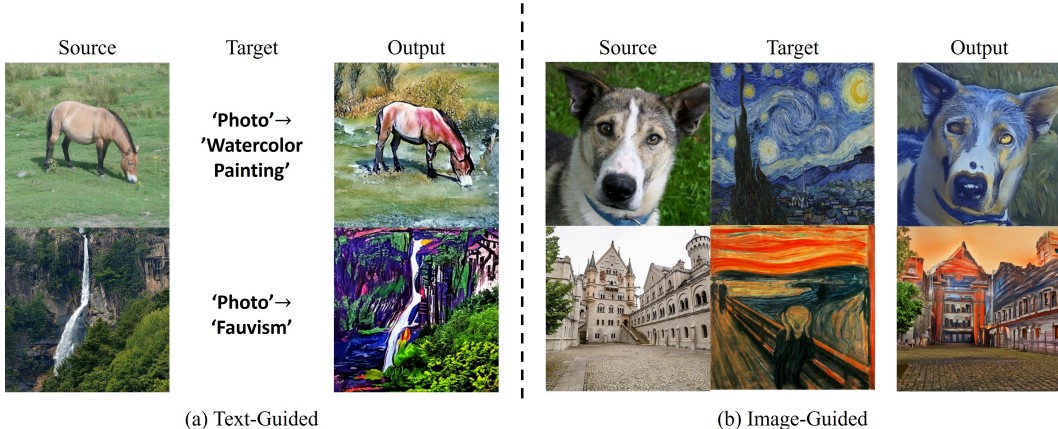

(a) Text-Guided                              (b) Image-Guided

Figure 7: Various outputs of artistic style transfer. We can translation natural images into artistic style paintings with both of text or image conditions.

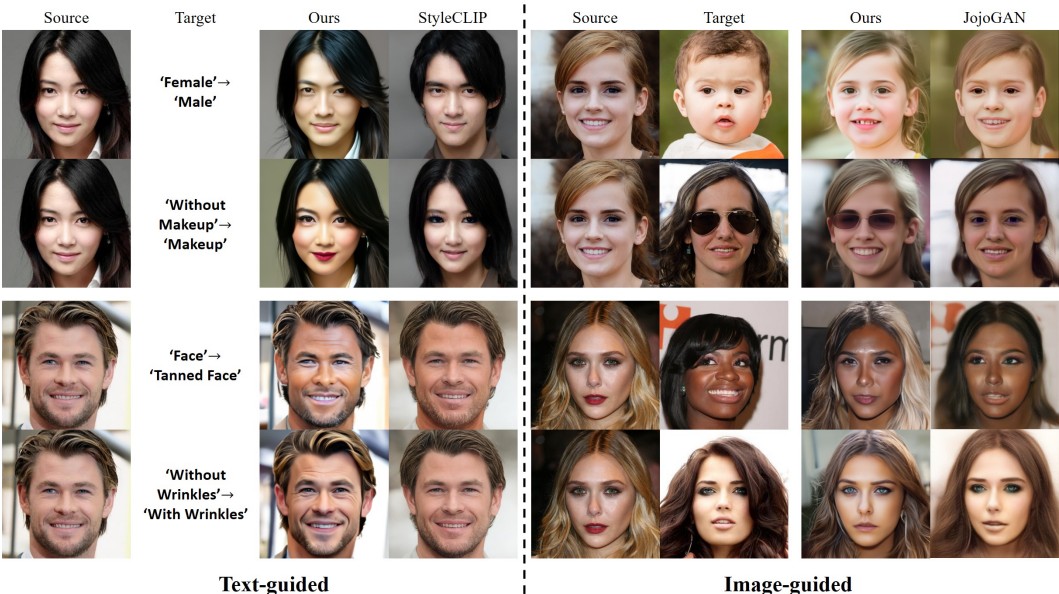

Figure 8: Outputs from face image translation models. The outputs from our model successfully translated the human face images with proper target domain semantic information.

In Fig. 8, we show that our method also can be used in face image translation tasks. For comparison, we included baseline models of face editing method StyleCLIP (Patashnik et al., 2021), and one-shot face stylization model of JojoGAN (Chong & Forsyth, 2021). The results show that our method can translate the source faces into target domain with proper semantic change. In baseline models, although some images show high quality outputs, in most cases the image failed in translating the images. Also, since the baseline models rely on pre-trained StyleGAN, they require additional GAN inversion process to translate the source image. Therefore, the content information is not perfectly matched to the source image due to the limitation of GAN inversion methods.

## E INFERENCE TIME COMPARISON

To evaluate the time-efficiency of our method, we calculate the inference times of the various image-guided translation models. All experiments are conducted with single RTX3090 GPU, on the same

| | Ours | SplicingVit | STROTSS | WCT2 | SANET |
|---|---|---|---|---|---|
| time | 37s | 25m 30s | 53s | 0.18s | 0.12s |

Table 4: Quantitative comparison on inference times of image-guided translation models.

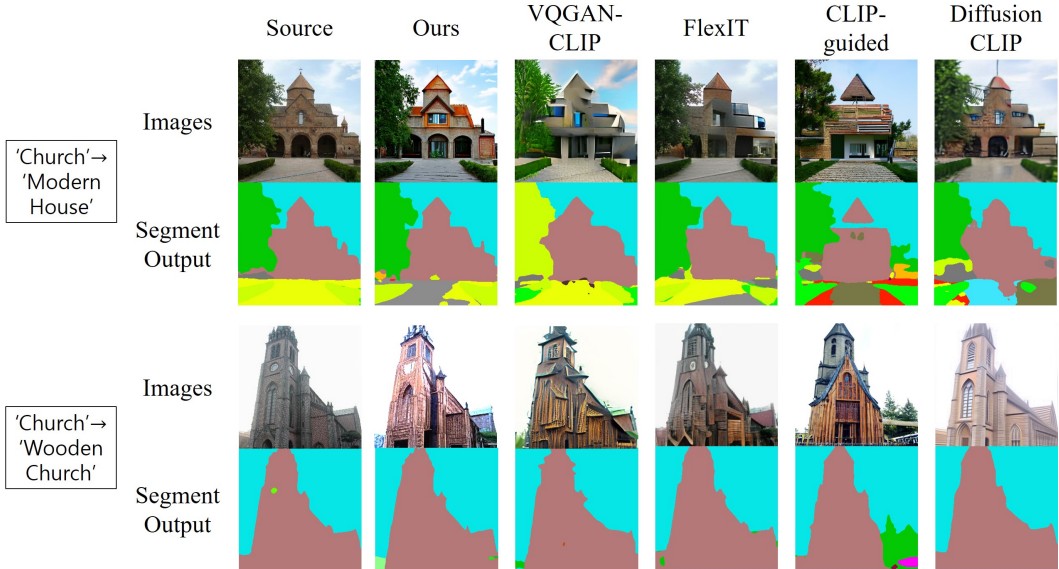

Figure 9: Comparison results on semantic segmentation maps from baseline outputs. When comparing segmentation maps, our model outputs show high structural consistency with the source images.

hardware and software environment. We use the images of resolution 256×256 for experiments. In Table 4, we compare the times taken for single image translation. For single-shot semantic transfer models of Splicing ViT, the inference time is relatively long as we need to optimize large U-Net model for each image translation. In STROTSS, it requires texture matching calculation for single image translation, so it takes long time. For arbitrary style transfer models of WCT2 and SANet, the inference is done with only single-step network forward process, as the model is already trained with large dataset. Our model takes about 40 seconds, which is moderate when compared to the one-shot semantic transfer models (SplicingVit,STROTSS). However, the time is still longer than the style transfer models, as our model need multiple reverse DDPM steps for inference. In the future work, we are planning to improve the inference time with leveraging recent approaches.

## F    SEMANTIC SEGMENTATION OUTPUTS

To further verify the structural consistency between output and source images, we compared the semantic segmentation maps from outputs and source images. For experiment, we use semantic segmentation model (Zhou et al., 2017) which is pre-trained on ADE20K dataset. We referenced the official source code[10] for segmentation model. Figure 9 shows the comparison results. In case of the baseline models VQGAN-CLIP, CLIP-guided diffusion, we can see that the segmentation maps are not properly aligned to the source maps, which means the model cannot keep the structure of source images. In case of FlexIT, the model outputs maps have high similarity to the source maps, but the semantic change is not properly applied. In our model, we can see the output maps have high similarity to the source maps, while semantic information is properly changed.

Source    Target    Ours    SANet    SANet+ViT

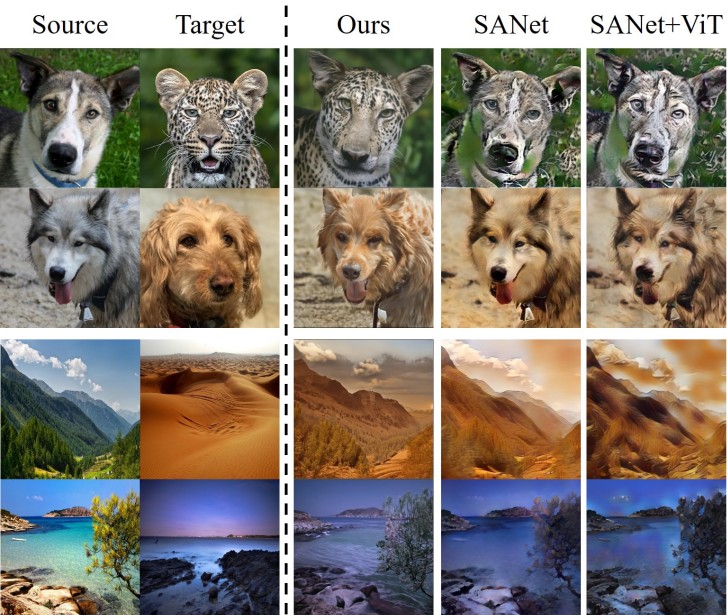

Figure 10: Additional comparison on image-guided translation. For fair experiment conditioning, we trained the baseline SANet with ViT-based losses.

Source    Target    Ours    Ours w/o L2 loss

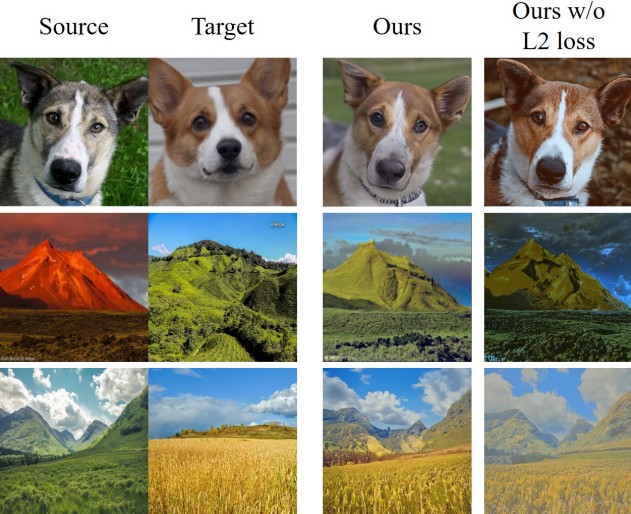

Figure 11: Ablation study results on pixel-wise l2 loss. Without pixel loss, the output image color is not matched to the color scale of the target images.

## G    ADDITIONAL COMPARISON ON IMAGE-GUIDED TRANSLATION

For fair comparison with the baseline models, we trained the baseline SANet with our proposed ViT-based loss functions. When we train SANet with replacing the existing style and content loss with our $\ell_{cont}$ and $\ell_{sty}$, we found that the training is not properly working. Therefore, we simultaneously used existing VGG-based style and content loss with our proposed ViT-based losses. In Fig. 10, we can see that when training SANet with ViT, the results still show incomplete semantic transfer results. Although the output seems to contain more complex textures than basic model, the model performance is still confined to simple color transformation.

---

[10]https://github.com/CSAILVision/semantic-segmentation-pytorch

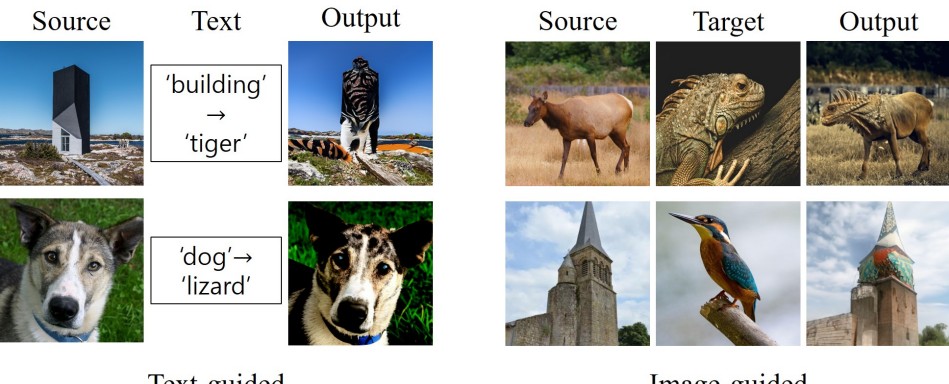

| Source | Text | Output | | Source | Target | Output |

Text-guided                                    Image-guided

Figure 12: Failure case outputs. If the semantic distance between source and target conditions are extremely far, semantic translation sometimes fails.

To further evaluate the effect of pixel-wixe l2 loss for image-guided translation, we conducted additional experiments in Fig. 11. When we remove the pixel-wise l2 loss in our image-guided translation task, we can see the semantic of output images follow the target images, but the overall color of the output images are slightly unaligned with the target image color. The result show that using weak l2 loss help the model to accurately apply the color of target images to outputs.

## H LIMITATION AND FUTURE WORK

Although our method has shown successful performance in image conversion, it still has limitations to solve. First, if the semantic distance between the source image and the target domain is too far (e.g building → Tiger), the output is not translated properly as shown in Fig. 12. We conjecture that this occurs when the text-image embedding space in the CLIP model is not accurately aligned, therefore it can be solved by using the advanced text-to-image embedding model. Second, our method has limitation that the image generation quality heavily relies on the performance of the pre-trained score model. This can also be solved if we use a diffusion model backbone with better performance. In future work, we plan to improve our proposed method in these two directions.

## I ADDITIONAL RESULTS

For additional results , in Fig. 13 we show the image translation outputs using text conditions. In Fig. 14, we additionally show the results from our image-guided image translation. We can successfully change the semantic of various natural images with text and image conditions.

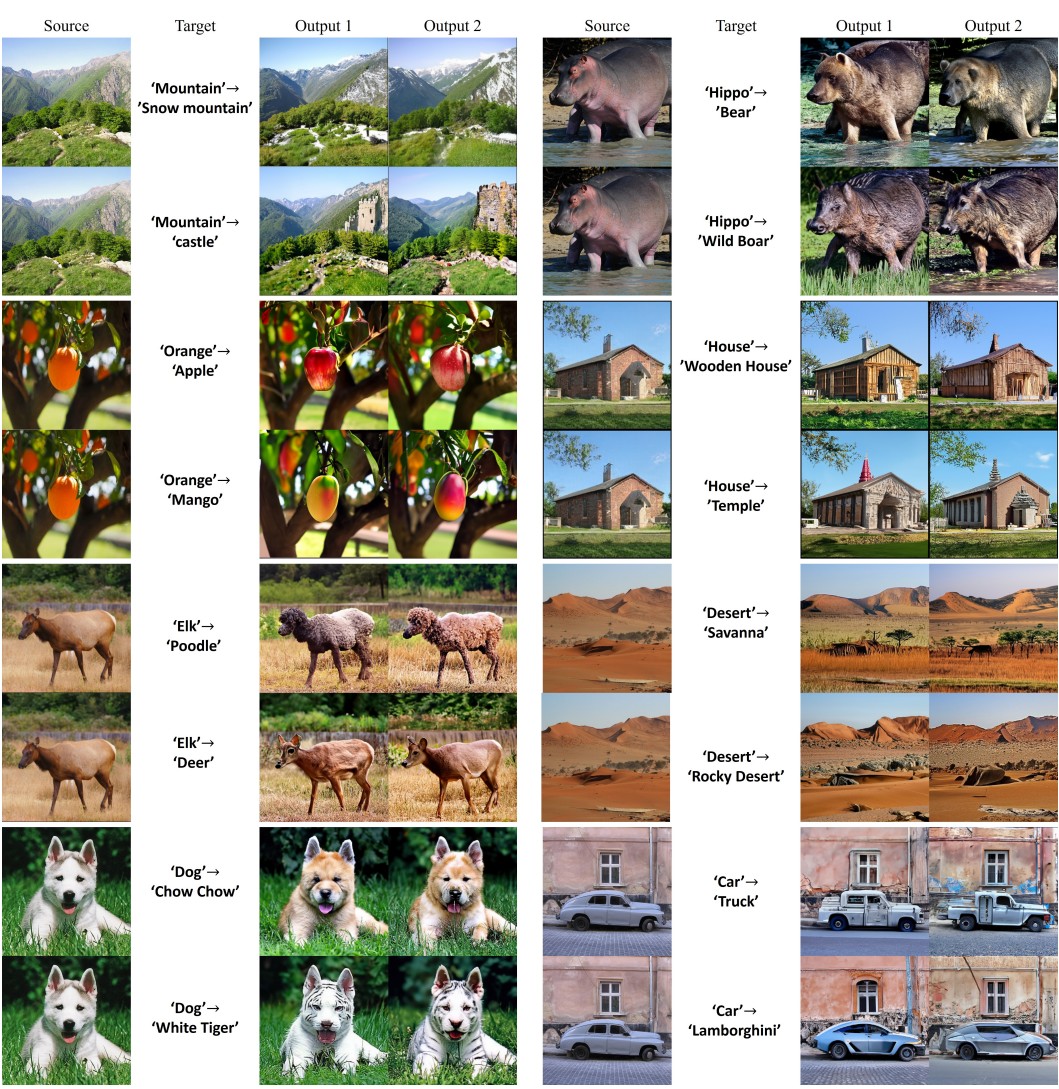

Figure 13: Qualitative results of text-guided image translation.

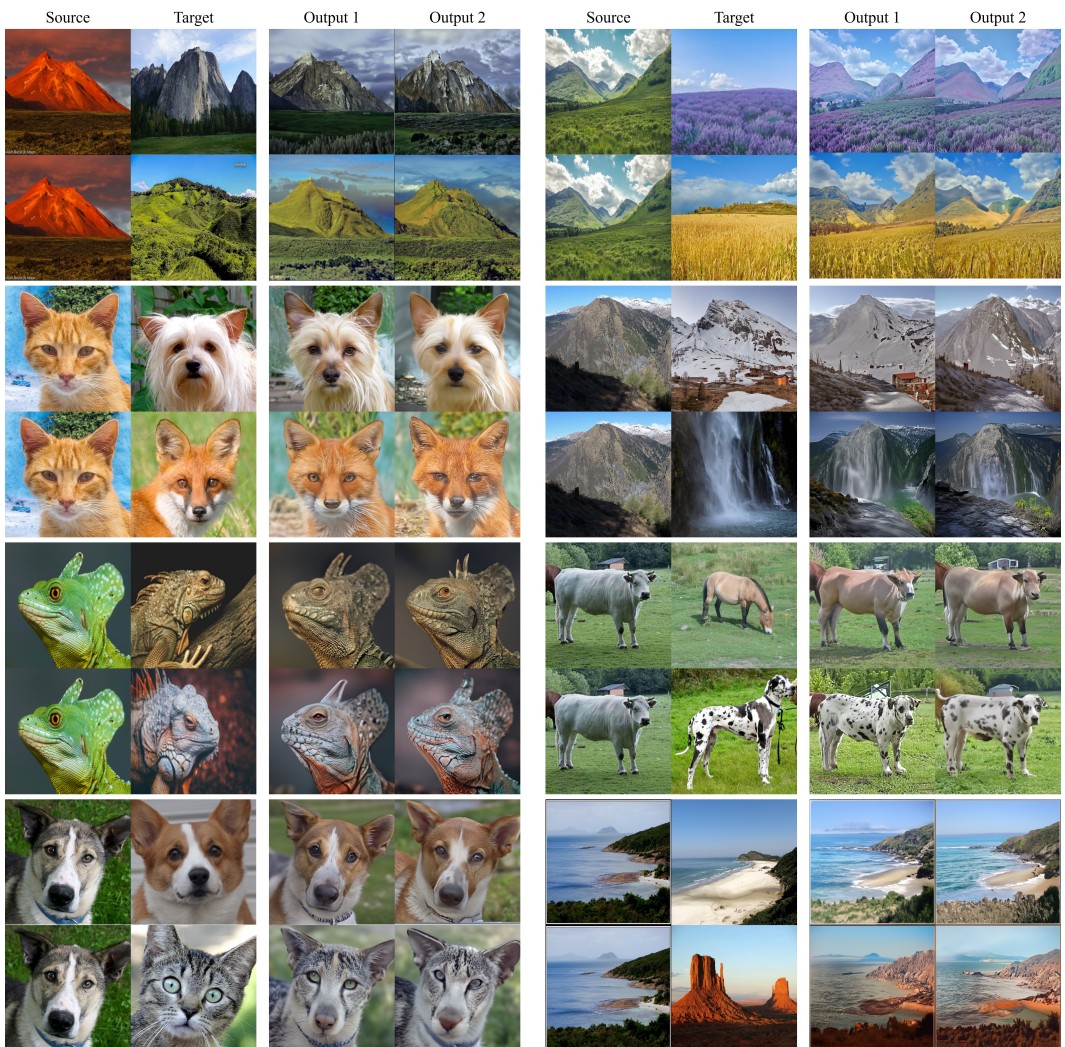

Figure 14: Qualitative results of image-guided image translation.

