# OpenReview forum: "Diffusion-based Image Translation using disentangled style and content representation"
_ICLR.cc/2023/Conference — ICLR 2023 poster_

### Official Review · Reviewer_CBU4 · 2022-10-23

**Confidence:** 3
**Correctness:** 4
**Technical Novelty And Significance:** 3
**Empirical Novelty And Significance:** 3
**Recommendation:** 8

**Clarity, Quality, Novelty And Reproducibility:**

The paper is well-written and easy to follow. The authors has conducted extensive experiments.

**Strength And Weaknesses:**

Strength:
1 The empirical results are pleasing. It outperforms all the existing methods.

2 The authors have conducted extensive experiments on Sec4.4 and the Appendix to support their ideas.

Weaknesses:

1. It seems that the contribution of this paper is to propose  L_cont and L_sem to represent the content and style information. However, it is not clear why such representation is superior compared with alternatives (e.g., VGG-based ones[1]). It will be interesting if the authors could have a thorough comparison.

2. there are some typos in the paper, e.g.,  in Fig. 2. The L_con should be L_cont. Please carefully proofread the paper.

[1] Perceptual Losses for Real-Time Style Transfer and Super-Resolution.

**Summary Of The Paper:**

This paper presents a diffusion-based image translation approach. It proposes a new way to disentangle the style and content representations. The content information is represented by the key similarity of a pretrained ViT model, and the  style information is represented by the last layer of CLS token.

**Summary Of The Review:**

Overall, the presented idea looks new, and the performance is promising. The authors have also conducted significant experiments to support their claims. Therefore, I vote for acceptance at this stage.

---

> ### Author Response · Authors · 2022-11-11
> **Thank you for the questions and the feedbacks.**
>
> **[W1]The reviewer requested ablation study with VGG pre-trained network.**
>
> Per the reviewer’s request, we carried out ablation study on VGG-based loss function. Instead of using key components of ViT in our model, we use features of VGG layer ( relu 3_1 ) for structural loss. Also, for semantic divergence loss, we extracted high layer feature( relu 5_1 ) instead of [CLS] token of ViT, as high layer features of VGG network is known for containing the semantic information.
>
> We included the ablation result in Figure 6(g). When VGG-based loss is used, we can see the structure of the output images are degraded. In Appendix Table 3, the quantitative results also show that the model shows decent scores in both SFID and CSFID, but the LPIPS scores is very high, which further confirms that it failed to preserve the structural information.
>
> **[W2] Minor comments about the typos.**
>
> We corrected the typos. Thank you for the feedbacks.

---

> ### Author Response · Authors · 2022-11-23
> **We are looking forward to your feedback.**
>
> Dear Reviewer CBU4,
>
> We thank you for your time and efforts in reviewing our paper, and your constructive comments. We would like to kindly ask you whether our reply and revised paper addressed your concerns. Could you please go over our responses and let us know if there are any remaining issues?
>
> Best regards,
>
> Authors

---

> > ### Comment · Reviewer_CBU4 · 2022-11-24
> > **Response**
> >
> > Thanks the authors for the response. They've addressed my concerns, and I don't have any further question.

---

### Official Review · Reviewer_9VMA · 2022-10-24

**Confidence:** 2
**Correctness:** 4
**Technical Novelty And Significance:** 3
**Empirical Novelty And Significance:** 3
**Recommendation:** 6

**Clarity, Quality, Novelty And Reproducibility:**

The idea is simple, and the methodology is described clearly, which makes it easy to follow. The experimental studies are sufficient. The novelty of this paper should be above the borderline.


**Strength And Weaknesses:**

Strength:

* The paper is well-written.
* The experiments are sufficient.
* The motivation for preserving the structure and changing the style is reasonable.

Weakness:
* The goal of keeping content structure unchanged can be verified by segmentation experiments rather than user studies.
For example, you can train a segmentation model on the target domain and then test it on the images generated from the source domain. Since the content structures are preserved, the predicted mask would be close to the mask of source images.


**Summary Of The Paper:**

This paper introduces a modified Denoising Diffusion Probabilistic Model with a proposed content preservation loss. It aims to achieve image-to-image translation with the content structure preserved and style changed. To do so, a content reservation loss and a resampling mechanism are proposed. Additionally, experiments of text and image-guided semantic image translation are established to verify the effectiveness of the proposed method.

**Summary Of The Review:**

This paper introduces a modified Denoising Diffusion Probabilistic Model with a proposed content preservation loss. It aims to achieve image-to-image translation with the content structure preserved and style changed, which is well-motivated. To do so, a content reservation loss and a resampling mechanism are proposed. The experimental studies seem to be sufficient. Thus, it might reach the publication demands.

---

> ### Author Response · Authors · 2022-11-11
> **Thank you for the questions and the feedbacks.**
>
> **[W1] The reviewer requested to show structural consistency with segmentation maps.**
>
> To further show the structural consistency, we included the visualizations using the segmentation maps in Figure. 9 and the explanations in Appendix F.  When comparing segmentation maps between ours and baselines, our model shows high consistency with source image segmentation maps with proper semantic change.

---

> ### Author Response · Authors · 2022-11-23
> **We are looking forward to your feedback.**
>
> Dear Reviewer 9VMA,
>
> We thank you for your time and efforts in reviewing our paper, and your constructive comments. We would like to kindly ask you whether our reply and revised paper addressed your concerns. Could you please go over our responses and let us know if there are any remaining issues?
>
> Best regards,
>
> Authors

---

### Official Review · Reviewer_swjc · 2022-10-25

**Confidence:** 4
**Correctness:** 4
**Technical Novelty And Significance:** 3
**Empirical Novelty And Significance:** 2
**Recommendation:** 6

**Clarity, Quality, Novelty And Reproducibility:**

*Clarity*: This paper is well-written.

*Quality and novelty*: The authors leverage guided diffusion for single shot image translation, and the visual results are more realistic compared to other methods.

*Reproducibility*: I believe the authors provide sufficient detail to reproduce the results in the paper.

**Strength And Weaknesses:**

*Strength*
1. The paper is well-written. The authors clearly introduce and detail the proposed approach.
2. The proposed method is shown to be more effective compared to existing guided diffusion methods on the text-guided image translation task.

*Weakness*
1. In common image-to-image translation, "style" referes to the variation which is not shared across different visual domains, which is actually defined in a data driven manner. For example, in AFHQ, style indicates the species, while in winter <> summer, style represents the color variation. Since the two losses (CLIP and semantic style) the authors use only encodes high-level semantic information, how does the proposed method handle the case that style encodes the low-level (e.g., color) variation? For example, the orange sky color is not correctly transfer to the output image in Figure 7 (b). Or more general speaking, how do the proposed method learns to differentiate what is common/unique variation that shared/not shared across different visual domains?
2. Although better visual quality is shown in the image guided translation compared to other methods, diffusion-based model is known for (relatively) long inference time. How long does it take for the proposed method to synthesize an image from an image guidance? How does that compare with other single-shot image translation method?

**Summary Of The Paper:**

This paper works on text-guided and image-guided image translations. The authors propose a guided diffusion method that tries to 1) maintain the structure of the source image, and 2) generate semantic content that matches the guidance. Specifically, contrastive loss, CLIP loss, semantic style loss and divergence loss is used during the reverse diffusion process. Experimental results demonstrate the effectiveness of the proposed method over several guided diffusion approaches and some image-to-image translation models.

**Summary Of The Review:**

Overall, the paper is well-written and the results are somewhat promising. I hope the authors can address the concerns raised in the Weakness section.

---

> ### Author Response · Authors · 2022-11-11
> **Thank you for the questions and the feedbacks.**
>
> **[W1] The reviewer asked further explanations about how the model translates different levels of semantics.**
>
> In text-guided image translation, since CLIP model is pre-trained with large number of data (~400M pairs), the model can cover wide range is image representations from low level to high level semantics. With this advantages, recent text-to-image generation models already leveraged the CLIP encoders to generated images from low- to high-level semantics. Low level or high level is expressed in different degrees in CLIP depending on which text condition is used. For example, when we use text condition ‘photo’ -> ‘painting’ it transfer low level texture or color information, and in case of using ‘lion’ -> ‘tiger’ , it transfer relatively high level semantics (species).
>
> In Image-guided translation tasks, DINO ViT is also pre-trained with large data so it can cover from low level to high level semantics. However, we observed that only using [CLS] has limitation in transferring low level semantics (e.g. color). Therefore, we assisted the low level transformation with weak pixel-wise l2 loss.
>
> **[W2] The reviewer requested to compare inference time for image-guided models.**
>
> Per the reviewer’s request, we compare the inference times between ours and baseline methods in Appendix Table 4. In semantic texture transfer models (Splicing Vit, STROTSS), they take longer time as they require optimization step for each translated images. In arbitrary style transfer models (WCT,SANet), inference time is level of miliseconds, as they only require one network forward step.
>
> In our method, the generation takes about 40 seconds. Considering the output quality, it is moderate time compared to the state-of-the-art semantic transfer models. However, the time is still longer than the style transfer models, as our model need multiple reverse DDPM steps for inference. In the future work, we are planning to improve the inference time with leveraging recent approaches (e.g. latent diffusion).

---

> > ### Comment · Reviewer_swjc · 2022-11-23
> > **Responses to the author feebacks**
> >
> > Thanks for the responses, especially about the inference time analysis.
> > For image-guided translation, could the authors show some low-level translation results WITHOUT using the weak pixel-wise l2 loss? This would clearly suggest the role of the pixel-wise l2 loss on this task.

---

> > > ### Author Response · Authors · 2022-11-23
> > > **Thank you for your additional comment.**
> > >
> > > Thank you for your comment. To answer your question, we conducted additional experiments.
> > > Since we cannot upload the revised paper now, we show the generated images in our anonymous drive.
> > >
> > > https://drive.google.com/file/d/1aZF52TsNqUs1148DtUR9dxziRH4wiS4d/view?usp=share_link
> > >
> > > When we remove the pixel-wise l2 loss in our image-guided translation task, we can see the semantic of output images follow the target images, but the overall color of the output images are slightly unaligned with the target image color.
> > >
> > > The result show that using weak l2 loss help the model to accurately apply the color of target images to outputs.
> > >
> > > We will include the comparison result in our final version.

---

> > > > ### Comment · Reviewer_swjc · 2022-11-23
> > > > **Thank you for the feedbacks**
> > > >
> > > > Thank you for the responses and the great work! I hope this work can further stimulate the research on diffusion models.

---

> ### Author Response · Authors · 2022-11-23
> **We are looking forward to your feedback.**
>
> Dear Reviewer swjc,
>
> We thank the reviewer for your time and efforts in reviewing our paper, and your constructive comments. We would like to kindly ask you whether our reply and revised paper addressed your concerns. Could you please go over our responses and let us know if there are any remaining issues?
>
> Best regards,
>
> Authors

---

### Official Review · Reviewer_VSVg · 2022-10-26

**Confidence:** 5
**Correctness:** 3
**Technical Novelty And Significance:** 2
**Empirical Novelty And Significance:** 2
**Recommendation:** 6

**Clarity, Quality, Novelty And Reproducibility:**

The paper is clearly written with a good amount of details.
While the results look impressive, the novelty is a bit limited as mentioned in the review.
The paper contains sufficient details for a domain expert to reproduce. However, it would be great if the code can be open-sourced in the future.


**Strength And Weaknesses:**

Strengths:
* [S1] This paper presents a unified framework to improve guided image translation with a pre-trained diffusion model. Results on both text-guided and image-guided applications look more realistic (in terms of shape and texture) compared to the existing baselines, including very recent ones published this year.
* [S2] The combination of different techniques including slicing Vision Transformer, Manifold Constrained Gradient (MCG), and Come-closer-diffuse-faster (CCDF) make a lot of sense. It is great to see a combination of recent techniques really make a big difference on the guided image translation quality.

Weaknesses:
* [W1] Although results are very impressive on the animals and landscapes dataset, the technical novelty of the paper is very constrained. In the Figure 2, almost all the components have been explored in the previous work. To be fair, this paper is the first one that shows great results by combining all of them in a unified framework.

* [W2] As this paper leverages many pre-trained models such as a pre-trained image diffusion model, a ViT model, and a CLIP model, the reviewer feels the comparison to existing baselines in the guided image translation setting is a bit unfair. For example, what happens if we leverage pre-trained models and apply loss functions to aid the SANet (Park and Lee, 2019). It would be good to show some fair qualitative comparisons in the image translation setting (in addition to the Section 4.4).

* [W3] It is questionable if the guided translation results look very diverse or not. The reviewer would like to know if this is a sign of memorizing the training set (from the pre-trained models). For example, it would be good to find the closest example in the ImageNet and show them on the side. Alternatively, it would be good to show multiple translation results from either image-guided and text-guided applications.

* [W4] The reviewer noticed that FlexIT actually achieved better LPIPS score in Table 1. It is true that FlexIT is trained directly with LPIPS score, as explained in the paper. The reviewer still feels that the proposed model is structure-aware (due to disentanglement) but not very content-preserving. It would be good to elaborate a bit on this.

* [W5] What are the failure cases of the proposed method? It is important to mention the failure cases in the main text and show more results in the supplementary material.

**Summary Of The Paper:**

This paper studies the problem of (text/image) guided (single) image editing using a pre-trained diffusion model. Specifically, it proposed a novel diffusion-based image translation method by disentangling style and content representations. Borrowed from the disentangling technique introduced recently (Tumanyan et al., 2022), a pre-trained Vision Transformer (ViT) has been used to aid the style and content disentanglement during the diffusion process. To accelerate the reverse diffusion process, this paper also applied two techniques, namely, semantic divergence loss and resampling strategy (Chung et al., 2022b). Experimental evaluations have been conducted on Animals and Landscapes datasets. Both quantitative and qualitative results demonstrate the strength of the proposed method over existing baselines such as Splicing ViT (Tumanyan  et al., 2022), VQGAN-CLIP (Crowson et al., 2022), and Flexit (Couairon et al., 2022).


**Summary Of The Review:**

Overall, I think this is a good paper. I am happy to raise my score if the concerns can be addressed in the rebuttal.

---

> ### Author Response · Authors · 2022-11-11
> **Thank you for the questions and the feedbacks.**
>
> **[W1] Reviewer has concerns about technical novelty due to the combination of the existing components.**
>
> Contrary to the concerns of the reviewer, we would like to assure the reviewer that the exploration of novel combinations of pre-trained models is not a weakness but an important innovation of this work given the advances in the foundation models (i.e. high-performance pre-trained models) in recent years. The reviewer is kindly reminded that many high-performance, large-scale pre-trained foundation models are now publicly available. For example, the current prompt learning, which optimizes the text or visual prompt instead of refining the foundation model, is a new research trend, and it is believed that a new machine learning trend would be more to find an optimal combination of the pretrained foundation models without too much relying on a new neural network training in an end-to-end manner. Our work is therefore the first of its kind that represents an optimal combination of the existing pre-trained model for a new application target that is not possible from individual models. Aside from these philosophical benefits, the reviewer is kindly reminded that there are also many important new contributions in our work. For example,
>
> * Our proposed structure loss  $l_{cont}$ is novel, which has not been attempted in the previous work. It is first trial to apply contrastive learning to extracted key features of ViT model. The effectiveness of our proposed loss can be confirmed from our ablation study.
>
> * Combining pre-trained ViT with Diffusion framework is the very first trial.
>
> * In acceleration strategy, our work is the very first work that can accelerate the diffusion process with semantic feature maximization between previous and present step outputs. Also, our resampling strategy is novel, as we alternated reverse gradient guided diffusion step and forward noising step in the last timestep t=T. The proposed resampling strategy has never been attempted in the previous methods.
>
> * Our method is so flexible that it can successfully translated the source image with both of text condition and image condition. There are several diffusion-based approaches which can generate and manipulate images with text conditions, but our framework is the very first work which can translate images using both of text and image. Our experimental results show that our method show state-of-the-art performance in both of text- and image- guided translation.
>
> **[W2] The reviewer requested comparison experiment with fair conditions.**
>
> Per the reviewer’s request., we show additional result in Figure 10. We trained SANet with our proposed $l_{cont}$ and $l_{sty}$. We empirically observed when we totally substitute the existing loss of SANET with our proposed loss, the model is not properly trained. Therefore, we simultaneously used existing SANet loss and our proposed ViT-based loss. In the result, although the models is assisted by ViT model, the model cannot change the entire semantic of input image, still concentrating on color components.
>
> **[W3] The reviewer requested to show diverse translation results.**
>
> The reviewer is kindly reminded that we already showed diverse outputs in our Appendix, Fig 12 and Fig 13. The result show that our method is not just generating the memorized samples.
>
> **[W4] The reviewer requested to elaborate more about the meaning of ‘structure’ and ‘content’. Also the reviewer requested explanation about LPIPS score of FlexIT.**
>
> In this paper, we defined ‘style’ as semantic information (e.g. color, species, texture), and ‘content’ as structural information (e.g. shape, component location). This terminology is borrowed from previous image translation work such as StarGAN, StarGANv2, MUNIT, and so on. We recognize that it would be confusing as FlexIT defined ‘content’ as semantic information, but eventually the goal of both models are the same : changing the semantic information, while preserving the structure of input image.
>
> In Flexit, they used LPIPS loss for structural consistency. However, LPIPS is a measurement of perceptual distance between the two images, not just for structural information. Therefore, it is not a perfect metric for measuring only structural consistency. Specifically, too low LPIPS means that it is not properly changed from source image, and too high LPIPS means that the structure consistency is severely degraded.
>
> To further show the structural consistency, we included the visualization using segmentation map in Fig. 9 and explanations in Appendix F.  When comparing segmentation maps between ours and baselines, our model shows high consistency with source image segmentation maps with proper semantic change.
>
> **[W5] The reviewer requested to show failure cases.**
>
> Per the reviewer’s request, we included failure cases in Appendix, Figure 11. With text conditions (or image conditions) which are too far from the source domain, the model cannot change the semantic of source.

---

> ### Author Response · Authors · 2022-11-23
> **We are looking forward to your feedback.**
>
> Dear Reviewer VSVg,
>
> We thank the reviewer for your time and efforts in reviewing our paper, and your constructive comments. We would like to kindly ask you whether our reply and revised paper addressed your concerns. Could you please go over our responses and let us know if there are any remaining issues?
>
> Best regards,
>
> Authors

---

### Author Response · Authors · 2022-11-13
**General comments by Paper 5649 authors**

We thank the reviewer for all of the comments and the reviews. Per all the reviewer’s request, we included several changes in our revised paper:

* We included additional comparison with SAnet trained with our proposed ViT –base loss functions in Appendix G, Figure 10.

* We included failure cases in Appendix Figure 11.

* In Appendix E - Table 4, we additionally show the inference time comparison results for image-guided translation case.

* For further evaluate the structural consistency performance, we show the qualitative comparison results with showing the segmentation map outputs from source and output images in Appendix F, Figure 9.

* For additional ablation study, we show the qualitative and quantitative results with using VGG-based loss functions instead of our proposed ViT losses. The qualitative output is shown in Figure 6(g), and the quantitative result is in Appendix B, Table 3.

* We corrected the typos and errors.

More details and explanation on the questions are elaborated in the following responses.

---

### Decision · Program_Chairs · 2023-01-20

**Decision:**

Accept: poster

**Justification For Why Not Higher Score:**

The paper tackles a relevant problem, appears well executed, outperforms the baselines, and shows great qualitative results, which could motivate additional research. The novelty of the paper is however perceived as limited as the paper leverages previously explored techniques.

**Justification For Why Not Lower Score:**

The paper tackles a relevant problem, appears well executed, outperforms the baselines, and shows great qualitative results, which could motivate additional research.

**Metareview: Summary, Strengths And Weaknesses:**

This paper was reviewed by four knowledgeable referees. The reviewers found the contributions sound (VSVg) and simple (9VMA), the paper clearly written (VSVg, swjc, 9VMA), the qualitative results impressive/pleasing (VSVg, CBU), and the experiments rather convincing (9VMA, swjc, CBU4). The reviewers raised concerns w.r.t. the novelty (VSVg, 9VMA), the comparisons with methods which do not use pre-trained models (VSVg), the missing comparisons with losses based on different representations (CBU4), and the limitations (which were not discussed) (VSVg). The authors engaged in discussion with the reviewers and adequately addressed all their concerns. After discussion, some reviewers still find the novelty limited (combination of existing techniques) but they do agree that this paper is the first to show great results with those techniques, and has the potential to stimulate further research in this direction. Therefore, the reviewers lean towards acceptance. The AC agrees with the reviewers and recommends to accept. The AC strongly encourages the authors to include the reviewers feedback in the final version of their manuscript - in particular, toning down statements w.r.t. achieved diversity, including the limitations (failure cases) discussion in the main body of the paper, and adding the reported results without the pixel-wise l2 loss.

**Note From Pc:**

if the above contains the word "oral" or "spotlight" please see: "oral" presentation means -> notable-top-5% and "spotlight" means -> notable-top-25%. As stated in our emails, we are disassociating presentation type from AC recommendations